# The Application of MicroRNAs in Glaucoma Research: A Bibliometric and Visualized Analysis

**DOI:** 10.3390/ijms242015377

**Published:** 2023-10-19

**Authors:** Ruqi Zhang, Yuanyuan Tao, Jufang Huang

**Affiliations:** Department of Anatomy and Neurobiology, School of Basic Medical Sciences, Central South University, Changsha 410013, China; zhangruqi@csu.edu.cn (R.Z.); taoyuanyuan1021@csu.edu.cn (Y.T.)

**Keywords:** microRNAs, glaucoma, bibliometric analysis, VOSviewer, CiteSpace, collaboration network

## Abstract

Glaucoma is similar to a neurodegenerative disorder and leads to global irreversible loss of vision. Despite extensive research, the pathophysiological mechanisms of glaucoma remain unclear, and no complete cure has yet been identified for glaucoma. Recent studies have shown that microRNAs can serve as diagnostic biomarkers or therapeutic targets for glaucoma; however, there are few bibliometric studies that focus on using microRNAs in glaucoma research. Here, we have adopted a bibliometric analysis in the field of microRNAs in glaucoma research to manifest the current tendencies and research hotspots and to present a visual map of the past and emerging tendencies in this field. In this study, we retrieved publications in the Web of Science database that centered on this field between 2007 and 2022. Next, we used VOSviewer, CiteSpace, Scimago Graphica, and Microsoft Excel to present visual representations of a co-occurrence analysis, co-citation analysis, tendencies, hotspots, and the contributions of authors, institutions, journals, and countries/regions. The United States was the main contributor. *Investigative Ophthalmology and Visual Science* has published the most articles in this field. Over the past 15 years, there has been exponential growth in the number of publications and citations in this field across various countries, organizations, and authors. Thus, this study illustrates the current trends, hotspots, and emerging frontiers and provides new insight and guidance for searching for new diagnostic biomarkers and clinical trials for glaucoma in the future. Furthermore, international collaborations can also be used to broaden and deepen the field of microRNAs in glaucoma research.

## 1. Introduction

Glaucoma is a neurodegenerative disorder that affects the visual system and eventually leads to a global and irreversible loss of vision [1]. Glaucoma currently affects millions of people worldwide and is expected to affect 111.8 million people by 2040 [2]. The slow loss of retinal ganglion cells (RGCs) and their axons is a characteristic symptom of glaucoma [3]. Moreover, despite considerable research, the mechanisms underlying the pathogenesis of glaucoma currently remain unclear, although an increase in intraocular pressure (IOP) is known to be the main risk factor for glaucoma [4]. Presently, a reduction in the intraocular pressure in glaucoma patients is the only verified treatment approach, including for normal IOP glaucoma patients [5,6]. Glaucoma can be classified into either open-angle glaucoma (OAG) or closed-angle glaucoma (CAG), which are further divided into primary and secondary [7], with primary open-angle glaucoma (POAG) being the most common type of glaucoma. Currently, there is no complete cure for glaucoma, with current treatments concentrating on delaying the progression of the disease, meaning that early diagnosis and treatment are essential [8,9]; however, the early detection of POAG remains unsatisfactory, which exacerbates the encumbrance of worldwide glaucoma.

Numerous research studies have shown that microRNAs (miRNAs) play an essential role in the pathogenesis of POAG [4]. Moreover, some studies have identified miRNAs as both diagnostic and prospective biomarkers for various diseases, such as cardiovascular diseases and neurodegenerative diseases, including retinal disorders and cancers [10,11,12]. miRNAs, which belong to endogenous non-coding RNA, are formed from small single-stranded nucleotides (~22 bp in length) [13], and recognize and bind to specific target mRNA sequences to either induce their degradation [14] or suppress their translation [15], thereby modulating the expressions of target genes through post-transcriptional regulation [16]. Previous research has shown that miRNAs are dysregulated in multiple diseases and may be involved in the pathogenesis of glaucoma [17]. Thus, various miRNAs were found to serve as biomarkers of POAG and aid in its early diagnosis [18]. For instance, it has been shown that miR-24, miR-29b, miR-200c, and miR-204 are potential diagnostic biomarkers or therapy targets for glaucoma [17,19]. Additionally, several miRNAs (e.g., miR-143-3p, miR-125b-5p, and miR-1260b) found in aqueous humor (AH) may serve as targets for therapeutic intervention [4]. Further, a gene mutation in miR-182 caused an increase in IOP and was found to be upregulated in the AH of POAG patients [20]. Overall, miRNAs play an important role in the diagnosis and treatment of glaucoma [21], and while there is a lot of potential for miRNAs in this area, challenges remain.

Bibliometric analysis is a commonly used scientific and quantitative research approach for publications, which includes collecting, processing, and managing the data from previous scientific publications to summarize the advancements in research topics and analyze the contributions of authors, institutions, journals, and countries or regions [22,23]. In addition, bibliometric analysis can identify hotspots, emerging tendencies, and knowledge networks in a particular field [24]. However, there are minimal studies available that have applied a bibliometric analysis to the field of miRNAs in glaucoma.

Therefore, in this study, we performed a systematic bibliometric analysis of the literature for research including miRNAs and glaucoma between 2007 and 2022. Here, we analyze the number of annual publications, the contributions of the authors, institutions, journals, and countries or regions, the publishing trends, the international collaborations, the co-occurrence visualization analysis of keywords, and the references. Moreover, we provide an outlook on the recent progress in research over the past 15 years and identify the research hotspots and trends in this field. Taken together, we aimed to summarize the past research and provide a foundation of research or novel frontier to enable further research to be performed on miRNAs in glaucoma.

## 2. Results

### 2.1. General Data

We initially retrieved a total of 209 publications from the Web of Science Core Collection (WoSCC) database, covering the period from 1 January 2007 to 28 December 2022. Next, the publication types were limited to original and review articles, and the publication language was limited to English. Finally, 184 publications were included in this analysis. The analysis process and items are shown in Figure 1. In accordance with the annual number of publications, the search period was divided into two stages (Figure 2A). In the first stage (2007–2014), the number of annual publications was less than 10, while in the second stage (2014–2022), the number of annual publications was greater than 10. The peak year was 2021. A total of 132 articles were published in the previous five years, which accounted for approximately 71.74% of the included publications. There were 152 (82.61%) original articles and 32 (17.40%) review articles included in our analysis. The overall number of citations was 4014, with an average of 21.96 citations per paper. Moreover, the following bibliometric parameters were determined: 29 countries or regions, 276 organizations, 96 journals, 1467 co-cited journals, 860 authors, 6646 co-cited authors, 1040 keywords, and 8104 references. The countries that published the most articles included China, the United States, and Iran (Figure 2B). Since 2017, the number of annual publications in China has grown rapidly, and this indicator in the United States and Iran has maintained a relatively stable growth.

### 2.2. Active Countries or Regions

There was a total of 29 countries/regions that had published in this research field; however, when the minimum number of documents was set to five, there were only six countries/regions that met the threshold. As shown in Figure 3A, we summarized the number of publications, total citations, and average citations in the top 10 prolific countries/regions. The most productive country was China (*n* = 104; 1199 citations; average of 11.5 citations), which accounted for 56.52% of the total included papers, followed by the United States (*n* = 42; 1590 citations; average of 37.9 citations). Iran was ranked third (*n* = 9; 1110 citations; average of 12.3 citations). However, it is worth noting that Germany, which ranked 6th in the number of publications among the top 10 prolific countries/regions, had the highest number of average citations. In this field, 27 countries had cooperative relations (Figure 3B). The United States was the most active country in the field and cooperated with nine countries.

### 2.3. Active Organizations

There were 276 organizations, of which seven organizations published more than five papers. Figure 4A shows the top 10 productive institutions, while their publications accounted for 38.04% of the included data. Duke University was the most prominent contributor in this field, with the highest number of publications and total citations. Fudan University ranked second. Of the top 10 productive institutions, six were based in China, three were in the United States, and the remaining one was located in Iran. A total of 57 organizations were connected, and we have constructed a map of their cooperative relationships (Figure 4B). The link strength between Duke University and Augusta University was the highest. In terms of average publication years, Harvard University had the earliest average publication year, with an index of 2013.67, while Dalian Medical University and Zhejiang University were the most recently active institutes in this domain with indexes of 2022 and 2021, respectively.

### 2.4. Top 10 Prolific Journals and Co-Cited Journals

We identified 96 journals using VOSviewer, while 18 journals met the threshold of a minimum of three publications. Table 1 summarizes the top 10 productive journals and co-cited journals and lists their relevant information. After sorting the number of publications, the *Investigative Ophthalmology and Visual Science* journal topped the list with 23 publications, while *Experimental Eye Research* (*n* = 10) and *Scientific Reports* (*n* = 6) ranked second and third, respectively. In terms of citations, *Investigative Ophthalmology and Visual Science* (750 citations) also ranked first, followed by *Experimental Eye Research* (211 citations) and *Molecular Vision* (157 citations). Notably, *Biomedicine and Pharmacotherapy* had the highest impact factor. In terms of co-citations, *Investigative Ophthalmology and Visual Science* (1107 co-citations), *Experimental Eye Research* (414 co-citations), and *PloS One* (332 co-citations) were the top three co-cited journals.

### 2.5. Active Authors

In this study, we analyzed the top 10 prolific authors and co-cited authors, as shown in Table 2. Sorted by the number of publications, Pedro Gonzalez, Coralia Luna, and Xinghuai Sun had all published six articles in this field. However, David Lee Epstein’s citations and average citations were the highest among the top 10 prolific authors. In terms of co-citations, Coralia Luna, Harry A. Quigley, and Ben Mead formed the top three (Table 2). Among the 36 authors who had published more than three articles, 17 authors had collaborated with others. Initially, we visually analyzed the cooperative relationships between these 17 authors according to cluster (Figure 5A). There were four clusters in total. William Daniel Stamer and Xinghuai Sun frequently co-operated with other authors, while Xinghuai Sun had the biggest total link strength. Based on the average publication years, we created an overlay map (Figure 5B). David Lee Epstein authored the earliest publications, whereas Junyi Chen, Yuan Lei, and Chen Tan authored the most recent publications.

### 2.6. Analysis of Keywords

Next, we obtained the research hotpots and trends by analyzing the co-occurrence of keywords [24]. This bibliometric information was analyzed by VOSviewer as follows: there were 1040 keywords in total, of which 59 appeared a minimum of five times, with a total of five clusters (Figure 6A). Next, we initially analyzed the frequently used keywords and found they mainly included “glaucoma” (*n* = 105), “microRNA” (*n* = 74), “expression” (*n* = 57), “open-angle glaucoma” (*n* = 44), “apoptosis” (*n* = 39), “aqueous humor” (*n* = 33), and “retinal ganglion cell” (*n* = 32). The keywords above also reflected the major themes associated with the investigators. In our overlay visualization map (Figure 6B), the light shade represents the most recent phase, and the dark shade represents the early phase. The following keywords reflected the recent attention of scholars in this field: “circular RNA” and “autophagy”. In addition, CiteSpace was also used to identify the major topics by detecting burst keywords during a particular period. Figure 6C exhibits the top 10 keywords alongside the strongest citation bursts, of which “open-angle glaucoma” was the keyword with the longest burst duration, and the burst strength of “genome wide association” was the highest.

### 2.7. Cited Publications and References

Table 3 lists the top 10 cited publications; among these articles, eight are original research and two are reviews. Jonathan D. Ashwell et al. published a research article in *Current Biology* in 2007, which had the highest number of citations. It is entitled “Optineurin negatively regulates TNF alpha-induced NF-kappa B activation by competing with NEMO for ubiquitinated RIP” [25]. There is a review article entitled “MicroRNA dysregulation in neurodegenerative diseases: A systematic review” [26], and although it was published in 2019, it ranked 4th in terms of citations. Furthermore, 2 of the top 10 cited articles were authored by Pedro Gonzalez. In addition, two articles were published in *Investigative Ophthalmology and Visual Science*. Based on CiteSpace, we obtained a graph spectrum of the highest cited references (Figure 7A) and the references with the strongest citation bursts (Figure 7B). In Figure 7A, the red nodes represent highly cited references, and the top five references are “Drewry MD (2018)” [27], “Hindle AG (2019)” [28], “Tanaka Y (2014)” [29], “Guo RR (2017)” [21], and “Dismuke WM (2015)” [18]. The first two references depict those with the longest burst durations (Figure 7B). Yuji Tanaka et al. published an original article titled “Profiles of Extracellular miRNAs in the Aqueous Humor of Glaucoma Patients Assessed with a Microarray System”, which had the highest burst strength [29]. There were three articles with citation bursts ending in 2022, meaning that these articles had received continuous focus in recent years.

**Table 3 ijms-24-15377-t003:** Top 10 most cited papers on the application of miRNAs in glaucoma research.

Rank	Title	Type	Citations	Journal	IF (2022)	Corresponding Author	Affiliation	Year
1	Optineurin negatively regulates TNF alpha-induced NF-kappa B activation by competing with NEMO for ubiquitinated RIP [25]	Article	212	*Current Biology*	9.2	Jonathan D. Ashwell	NIH National Cancer Institute	2007
2	Bone Marrow-Derived Mesenchymal Stem Cells-Derived Exosomes Promote Survival of Retinal Ganglion Cells Through miRNA-Dependent Mechanisms [30]	Article	206	*Stem Cells Translational Medicine*	6	Ben Mead	NIH National Eye Institute	2017
3	The role of TGF-beta in the pathogenesis of primary open-angle glaucoma [31]	Review	190	*Cell and Tissue Research*	3.6	Ernst R. Tamm	University of Regensburg	2012
4	MicroRNA dysregulation in neurodegenerative diseases: A systematic review [26]	Review	174	*Progress in Neurobiology*	6.7	Alyson E. Fournier	McGill University	2019
5	Role of miR-29b on the regulation of the extracellular matrix in human trabecular meshwork cells under chronic oxidative stress [32]	Article	132	*Molecular Vision*	2.2	Pedro Gonzalez	Duke University	2009
6	Human aqueous humor exosomes [18]	Article	79	*Experimental Eye Research*	3.4	Yutao Liu	University System of Georgia	2015
7	Coordinated Regulation of Extracellular Matrix Synthesis by the MicroRNA-29 Family in the Trabecular Meshwork [33]	Article	78	*Investigative Ophthalmology and Visual Science*	4.4	Douglas J. Rhee	Harvard University	2011
8	Profiles of Extracellular miRNAs in the Aqueous Humor of Glaucoma Patients Assessed with a Microarray System [29]	Article	72	*Scientific Reports*	4.6	Toru Nakazawa	Tohoku University	2014
9	MicroRNA-24 Regulates the Processing of Latent TGF beta 1 During Cyclic Mechanical Stress in Human Trabecular Meshwork Cells Through Direct Targeting of FURIN [34]	Article	70	*Journal of Cellular Physiology*	5.6	Pedro Gonzalez	Duke University	2011
10	Suppression of Type I Collagen Expression by miR-29b via PI3K, Akt, and Sp1 Pathway in Human Tenon’s Fibroblasts [35]	Article	67	*Investigative Ophthalmology and Visual Science*	4.4	Xuanchu Duan	Central South University	2012

**Figure 7 ijms-24-15377-f007:**
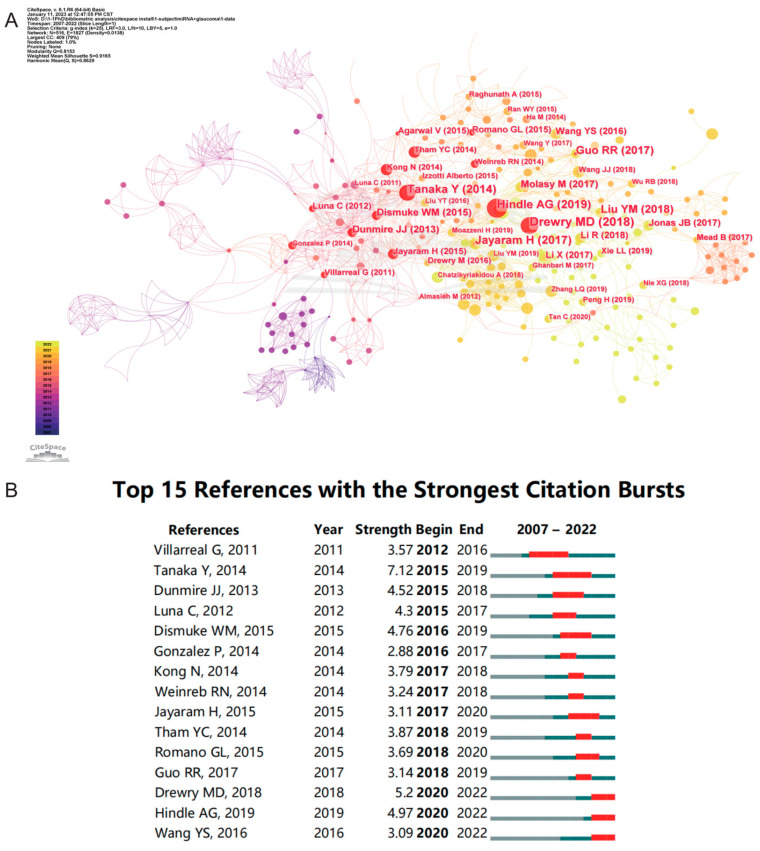
Analysis of references. (**A**) The highly cited references. (**B**) The top 15 references with the strongest citation bursts. The blue line represents the time from its first appearance to 2022; the red line represents the burst time [1,2,18,19,21,27,28,29,33,36,37,38,39,40,41].

## 3. Discussion

The number of annual publications can reflect the development of a particular field [42]. Thus, we identified an overall upward trend in this research area from 2007 to 2021; however, there was a decrease in the number of new publications in 2022. We speculate that this occurred for two reasons: firstly, it could be because the field has developed in other directions, or it could be due to a delay in the WoSCC collection. Our analysis included 184 articles, which had been published between 2007 and 2022, from a total of 29 countries/regions, 276 institutions, 96 journals, 1467 co-cited journals, 860 authors, 6646 co-cited authors, 1040 keywords, and 8104 references. These numbers indicate that this research field has received significant global attention. Indeed, the top three countries with the highest numbers of publications were China (*n* = 104), the United States (*n* = 42), and Iran (*n* = 9). In particular, China published a large number of articles after 2017, which may be linked to the intensification of the problem of population aging in China [43]. Of the top 10 prolific institutions, six were located in China, three were in the United States, and the remaining one was found in Iran. Duke University was the most prominent contributor and had the highest number of publications and total citations, while Fudan University ranked second. *Investigative Ophthalmology and Visual Science* (*n* = 23; 750 citations), *Experimental Eye Research* (*n* = 10; 211 citations), and *Scientific Reports* (*n* = 6, 115 citations) were the top three journals with the most publications in this field.

### 3.1. MicroRNAs

MicroRNAs are a class of endogenous single-stranded non-coding RNAs (ncRNAs) that are approximately 22 nucleotides in length [13] and regulate the expressions of over 60% of human protein-coding genes [44]. They were originally identified in *Caenorhabditis elegans* in 1993, while Lin-4 was the first miRNA to be identified by Ambros and Ruvkun [45,46]. Thereafter, vast volumes of miRNAs have been found in viruses, vegetation, and animals [47,48], with more than 2500 miRNAs having currently been determined in the human genome [49].

The biogenesis of miRNAs comprises the following steps [13]: (1) the microprocessor complex, which consists of DGCR8 recognizing pri-miRNAs, which are hundreds to thousands of nucleotides in length, and cleaves them into pre-miRNAs. (2) Ran-GTP and exportin-5 transport the pre-miRNA from the nucleus. (3) Dicer cleaves the pre-miRNAs into mature miRNAs. (4) Argonaute (AGO) proteins and mature miRNA form an RNA-induced silencing complex (RISC). (5) The target mRNA is specifically recognized and bound to by RISC before being cleaved.

While a few miRNAs function by recognizing individual targets, most of them act by incompletely binding to several targets, then inhibiting the translation of targets or inducing target degradation at the post-transcriptional level [50]. One target mRNA can be modulated by several miRNAs, and one miRNA can modulate several targets [51]. It is well known that miRNAs can command cell proliferation, differentiation, apoptosis, and metastasis [52,53,54], and can serve as regulators of multiple biological pathways [55,56]. Furthermore, miRNAs are also known to play key roles in a variety of physiological and disease processes by modulating whole signaling pathways. The deregulation of miRNAs in diseases can also facilitate the development of biomarkers [57] and provide some insight into potential disease pathogeneses [58].

### 3.2. MicroRNAs in Neurodegenerative Diseases (NDs)

miRNAs have been found to regulate gene expressions at the post-transcriptional level. Moreover, gene regulation has been found to be dysfunctional in multiple NDs and their animal models [59,60,61,62,63]. NDs are synonymous with the progressive deterioration of neuron structures and functions, ultimately resulting in neuronal loss, which is considered to be the basis of most neurological impairments [64]. Presently, NDs form one of the most severe health issues owing to the increase in aging populations. Unfortunately, they remain incurable and non-reversible even after many years of study. NDs, including Alzheimer’s disease, Parkinson’s disease, glaucoma, age-related macular degeneration, amyotrophic lateral sclerosis, Huntington’s disease, etc., usually have shared cellular mechanisms and histopathological characteristics beyond neuronal cell dysfunction or death [65]. Although the underlying causes of individual NDs are different, many common pathobiological characteristics and mechanisms are consistent in neurodegeneration across diseases. Recent studies have revealed that miRNA expressions are significantly altered in the pathogenesis of NDs and contribute to the physiology of abnormal neurons [60,66]. However, most of the literature reviews that depict the changes in miRNA expressions in specific NDs and the regulation of miRNAs across different NDs have not been systematically overviewed [67,68,69]. Thus, the identification of a common miRNA dysregulation processes across NDs may help us to better understand the conserved molecular pathways that are impacted in these diseases and highlight some new targets for treatment. Camille A et al. [26] determined the commonly dysregulated miRNAs across different NDs. Specifically, miR-9-5p, miR-21-5p, the miR-29 family, miR-124-3p, miR-132-3p, miR-146a-5p, miR-155-5p, and miR-223-3p are predominantly dysregulated in 12 classes of NDs and their representative animal models. Among them, miR-146a-5p, miR-155-5p, and miR-223-3p are mainly upregulated across different NDs. They also summarized the pathways that are targeted by the usually dysregulated miRNAs in NDs—for instance, Aβ genesis, autophagy, homeostasis, apoptosis, NF-κB signaling, etc. Moreover, they noted the functional overlap by these miRNAs and suggested that miRNAs may act synergistically to directly or indirectly act upon their targets.

### 3.3. MicroRNAs in Glaucoma

#### 3.3.1. Glaucoma

Glaucoma is a neurodegenerative disease, whereby a slow loss of RGCs and their axons will cause severe vision loss. As glaucoma progresses, the retinal nerve fiber layer and the optic nerve head suffer irreversible damage due to optic neuropathy [70]. To the best of our knowledge, the pathogenesis of glaucoma is still unknown; however, some risk factors of glaucoma have been identified, including advanced age, nearsightedness, race, IOP, reduced corneal thickness, positive family history of glaucoma, and potential vessel disease [71,72,73,74,75]. Some other factors, such as autoimmunity, defects in the autoregulation of ocular blood circulation, low intracranial pressure, aberrant structural sensitivity of the lamina cribrosa, and mitochondrial function disorders may also be engaged. Among these, IOP, which is caused by the blocked outflow of AH, forms the main risk factor [4]. Although the majority of patients possess an elevated IOP (>21 mmHg), there are also patients with low IOP values. The normal IOP values in healthy individuals range from 10 to 12 mmHg [76].

Glaucoma is categorized into OAG and CAG. OAG is the most frequent type of the disease, accounting for about 90% of all glaucoma patients [4,77], with POAG and exfoliation glaucoma, a kind of secondary glaucoma, representing the most common forms [78,79]. Among the many types of glaucoma, POAG is the most common, especially in people of European and African descent [80,81]. It is usually bilateral, although the severity is often asymmetric [2]. CAG is less common and is constituted in under 20% of patients in America. Primary angle closure glaucoma (PACG) is the most serious phase of closed-angle disorders [82].

Glaucoma is usually asymptomatic in its early stages of development and reducing the intraocular pressure is the only confirmed way to prevent or delay it. This kind of reduction can be achieved using non-surgical treatments, including eye drops and IOP-lowering medication, or surgical treatments, including laser trabeculoplasty, laser peripheral iridotomy, and incisional glaucoma surgery [7]. Although intraocular hypertension is frequently observed in patients with OAGs, only a small percentage of patients with ocular hypertension will develop this disease [83]. Despite the improved results from IOP-lowering treatments, a large number of glaucoma patients continue to lose their vision [84]. Predominantly, glaucoma is a chronic disease that requires lifelong therapy.

Numerous miRNAs are dysregulated in glaucoma and might play a vital role in the underlying pathogenic mechanisms of POAG [4]. Moreover, miRNAs can serve as diagnostic and prognostic biomarkers for glaucoma. For example, various miRNAs have been found in the patients-sourced samples with POAG [85], while they also fulfill a role in the trabecular meshwork (TM) [86], retina [87], and AH [29].

#### 3.3.2. MicroRNAs in Aqueous Humor (AH)

The main and sole known changeable risk factor of glaucoma is IOP, which is sustained through balancing AH production and the rate of AH discharge through the TM [88]. miRNAs derived from AH might play a critical role in the pathologic status associated with glaucoma, although full miRNA profiles have not yet been identified in the AH from glaucoma patients. The AH is an appealing source of neoteric miRNAs, which can serve as biomarkers or therapy targets in glaucoma, whereby its reachability, specificity, and separation is different from other organs [29]. We eagerly anticipate the identification of novel miRNAs that can diagnose the varying subtypes, pathologic conditions, and pharmacologic effects of glaucoma, which might be significantly expedited by the application of new analytical instruments. Overall, miR-143-3p, miR-125b-5p, and miR-1260b found in the AH might serve as promising therapeutic targets for glaucoma [4].

#### 3.3.3. MicroRNAs in Trabecular Meshwork (TM)

The damage associated with increased IOP is primarily characterized by the occurrence of TM degeneration [89]. TM is a vital constituent in the outflow pathway for AH and comprises the majority of the outflow resistance [90]. In POAG, a range of pathologic alterations is observed in the TM, especially in the elevation of extracellular matrix (ECM) molecules, such as collagens and fibronectins [91,92], which raise the outflow resistance and IOP [93]. Thus, it is necessary and urgent for us to clarify the pathologic mechanisms associated with the overproduction of ECM in the TM of glaucoma patients. To our knowledge, the synthesis of ECM is modulated by the transforming growth factor-β (TGF-β) family [94]. Additionally, studies have shown that the TGF-β/SMAD pathway can modulate ECM genes, which are related to increased IOP in glaucomatous optic neuropathy [95,96,97].

The diversely expressed miRNAs in the AH and blood of patients with glaucoma have previously been identified [27,98]. Recently, vast volumes of research have concentrated on the roles of miRNAs in regulating TM cell function during diverse pathologic status to attempt to identify a wider range of potential biomarkers for glaucoma therapy [99,100]. For example, the miR-29 family, which includes miR-29a, miR-29b-1, miR-29b-2, and miR-29c, plays an essential role in the development of fibrosis in POAG, owing to their antifibrotic effects on the TGF-β signaling pathway and the production and deposition of ECM [88]. Moreover, the miR-200 family, which includes miR-200a, miR-200b, miR-200c, miR-141, and miR-429, is a potential regulator of TM cell contractions. Among these, miR-200c has been shown to be the direct post-transcriptional inhibitor of genes associated with TM cell contraction, modulated trabecular contraction, and intraocular pressure in vivo [37]. Therefore, miR-200c is a valuable candidate for future therapeutic approaches in glaucoma research by regulating TM cell contraction. In addition, miR-486-5p can reduce the production of ECM and oxidative damage in human TM cells by targeting the TGF-β/SMAD2 pathway [101]. Overall, future miRNA-based therapies that focus on controlling the production of ECM, regulating TM cell contraction, or targeting the TGF-β may provide novel therapeutic approaches for glaucoma.

#### 3.3.4. Manipulating microRNA Expressions

A better comprehension of the effects of miRNAs in normal and abnormal eyes can help dissect the physiologic and pathologic mechanisms of glaucoma. The possibilities for miRNA-based therapies are enhanced by the capacities of miRNA manipulations to alter gene expressions both in vitro and in vivo [102,103]. miRNA-based therapies can be achieved as follows: (1) the application of miRNA mimetics. miRNA mimetics, which resemble miRNA precursors, can be used to downregulate certain target proteins; however, this approach may be accompanied by off-target effects [104]. (2) The application of anti-miRNAs. Anti-miRNAs can be used to promote cell survival and suppress cell apoptosis, which can inhibit endogenous miRNAs [105]. (3) The application of targeting the miRNAs’ processing mechanisms [17]. (4) Other applications, such as target site blockers and miRNA sponges [103].

#### 3.3.5. MicroRNAs in Mouse Models of Glaucoma

It is possible to explore the pathogenesis of human diseases using mouse models, which provide powerful research tools [106]. Regarding the significance of glaucoma mouse models for providing important mechanistic ideas to glaucoma pathogenesis, we analyzed and summarized the included articles that used glaucoma mouse models. In research that used acute glaucoma mouse models, in which the anterior chamber was cannulated with a needle, intravitreal infusion of polydopamine–polyethylenimine nanoparticles carried miR-21-5p [107], and mesenchymal stem cells (MSCs)-exosomes including miR-21a-5p [108,109] were used to provide new therapeutic pathways for neuroprotection against glaucoma. Moreover, there are some chronic glaucoma mouse models which were established by episcleral venous occlusion with cauterization [110], fixing a plastic ring to the ocular equator [111], infusing microbeads into the anterior chamber [112,113], laser photocoagulation [114], or directly using a genetic DBA/2J mouse model of glaucoma [115]. In these studies, the administration of microRNA mimic/inhibitor [110,111,112,114], application of microRNA knockout mice [113], and injection of bone marrow mesenchymal stem cell (BMSC)-derived exosomes including microRNAs [115] were used to develop microRNA-based advanced therapies. Furthermore, some other glaucoma models, such as optic nerve crush [116] and the injection of N-methyl-D-aspartic acid into the vitreous cavity [117], have also been applied in glaucoma research.

### 3.4. Other Non-Coding RNAs

miRNAs belong to endogenous ncRNAs, which form the greatest subset of RNA transcripts, which form 90% of the human genome. The ncRNAs do not translate into proteins and are involved in the regulation of diverse biological pathological processes [118,119]. Further, ncRNAs include miRNAs, circular RNAs (circRNAs), and long non-coding RNAs (lncRNAs).

#### 3.4.1. Circular RNAs

CircRNAs are a special class of ncRNAs that are highly expressed in mammalian cells and exhibit tissue specificity [120]. Moreover, they have been shown to be ideal biomarkers for human diseases [121,122].

CircRNAs can act as miRNA sponges, regulating downstream mRNA expressions, thus modulating cellular functions; they have been extensively accepted [123,124]. Some circRNAs can serve as gene regulators and engage in the regulation of a variety of biological processes [125]. The biofunction of circRNAs acts in three main ways: (1) circRNAs function as miRNA sponges; (2) circRNAs function as transcriptional regulators; (3) circRNAs function as biomarkers of disease progression [126]. There are various research studies that suggest that circRNAs are highly associated with the progress of diverse eye diseases [127,128], including glaucoma, age-related macular degeneration, retinal detachment, and diabetic retinopathy, and play a role in retinal dysfunction [129,130,131] by functioning mainly as sponges for microRNAs. For instance, it has been verified that hsa_circ_0023826 was downregulated in patients with glaucoma and can be used as a biomarker for glaucoma [132]. Furthermore, circZRANB1 is a possible candidate for glaucoma treatment due to its association with glaucoma-induced retinal neurodegeneration [133]. Currently, some candidate circRNAs have been screened by high-throughput sequencing technology, whereas the role of most circRNAs has not been completely elucidated in glaucoma. Yanxi Wang et al. [126] revealed that circ_0080940 might facilitate the advancement of glaucoma by sponging miR-139-5p; thus, the inhibition of circ_0080940 might be an attractive therapy for treating glaucoma. A single circRNA can act as a “sponge of miRNA” to disrupt the binding sites of several miRNAs and suppress the activation of one or more miRNAs [134]. The sponging of miRNAs by circRNAs in most disorders highlights their potential as therapy candidates or biomarkers. For example, silencing circZYG11B could suppress I/R-induced RGC injury and attenuate I/R-induced retinal reactive gliosis through circZYG11B/microRNA-620/PTEN signaling; thus, circZYG11B might be a candidate for the identification and therapy of retinal ischemic diseases [135]. Chen et al. [136] revealed that circHipk2 and circTulp4 can serve as sponges of miR-124-3p and miR-204-5p/miR-26a-5p, respectively, and the interruption of circTulp4 expression induced impaired retinal function. Additionally, circZNF609 can regulate retinal neurodegeneration by sponging miR-615, thereby silencing the circZNF609-induced suppression of retinal reactive gliosis and the activation of neurogliocytes to promote RGCs’ survival [131]. It is worth noting that the dysregulation of circRNAs in sponging miRNAs is what usually induces disease and not the circRNAs themselves. Moreover, evidence indicates that the dysregulation of the circRNAs might promote NDs, cardiovascular disorders, vascular diseases, and cancers [137,138].

#### 3.4.2. Long Non-Coding RNAs

The lncRNAs are a typical class of ncRNAs, are over 200 nucleotides in length, and are characterized by fewer exons and tissue or cell specificities [139]. LncRNAs are located mainly in the cell nucleus and can modify gene expressions at both the translational and transcriptional levels [140]. Moreover, lncRNAs can be subclassified based on the linkage to miRNAs and transcriptional direction [140]. Recent research has revealed that lncRNAs have multiple functions and play a modulating function in many ocular diseases, including glaucoma, diabetic retinopathy, age-related cataracts, and age-related macular degeneration [140]. In addition, some studies have shown that lncRNAs are closely related to the development of POAG [141,142]. Lili Xie et al. [142] suggested that lncRNAs T267384, ENST00000607393, and T342877 might be potential therapy biomarkers for POAG. Additionally, lncRNAs are prospective treatment candidates for preventing fibrosis after glaucoma filtration surgery [143]. However, the functions and associated mechanisms of the role most lncRNAs play in diseases have not been completely clarified.

#### 3.4.3. Competing Endogenous RNAs

Interestingly, lncRNAs may emerge as competing endogenous RNAs (ceRNAs) and can potentially correspond to specific messenger RNAs (mRNAs) through sponging to their target miRNAs [144,145,146]. Moreover, miRNAs can bind to their specific mRNAs and repress their expressions in the ceRNA network. Furthermore, lncRNAs can competitively bind to miRNA response elements (MREs) with mRNA and attenuate miRNA-mediated inhibition, while also mediating the post-transcriptional modulation of target genes to regulate cellular activities [147,148,149]. For instance, lncRNA TGFβ2-AS1 might facilitate ECM generation, which is associated with the development of POAG, by targeting TGF-β2 in human TM cells [150]. Similar to lncRNAs, circRNAs can also function as ceRNAs for miRNAs, whereby miRNAs play certain modulatory effects on mRNAs. For example, Zhichao Yan et al. [146] suggested that circXPO5 and GRIN2A can act as ceRNAs and compete with miR-330-5p to reduce circXPO5, which will have beneficial effects for glaucoma patients. CeRNAs are related to the molecular mechanisms in ocular disease, such as age-related macular degeneration [151]; thus, a ceRNAs network analysis can efficiently identify molecules related to their regulation. It has been revealed that the network of ceRNAs can contribute to the understanding of the potential molecular mechanisms involved in POAG progression. Additionally, some potential POAG biomarkers and bio-target molecules were identified using a ceRNA analysis [152,153]. The construction and identification of the lncRNA or circRNA–miRNA–gene ceRNA network may contribute to illustrating the pathogenesis of diseases and identifying prospective biomarkers for disorders [154].

### 3.5. MicroRNAs in Exosomes

Exosomes, which are a kind of MSCs-derived factor that contribute to the parasecretory effects of MSCs [155], can be used as a cell-free therapy with lower immunogenicity for treating retinal diseases [156] and inflammatory disorders [157]. Exosomes are part of the extracellular vesicles (EVs), which are secreted by viable cells and range from 30 nm to 150 nm in diameter [158]. Moreover, they include mRNAs, ncRNAs (miRNAs, circRNAs, and lncRNAs), proteins, transcription factors, and other biofactors, and have been verified as promising potential therapy options and drug deliverers [159]. It has been demonstrated that exosomes are involved in intercellular communication [160], and MSC-derived exosomes have been applied in various diseases, such as apoplexy [161], corneal diseases [162], glaucoma [115,163], and hepatopathy [164]. Growing evidence has shown that the biomarkers derived from exosomes may be helpful for diagnosing various disorders, including Parkinson’s disease [165] and diabetes mellitus [166].

Mead et al. demonstrated that the therapy effects of exosomes were at least partially attributable to their miRNAs in both the glaucoma rodent models [163] and genetic DBA/2J mouse models [115]. In the former study [163], they used two glaucoma rodent models: one with the induction of ocular hypertension with intracameral microbeads and the other with induction of ocular hypertension with laser photocoagulation. BMSC-derived exosomes were injected into the vitreous every week or every month, and the results show that the BMSC-derived exosomes provided significant neuroprotection to the RGCs while preserving the retinal nerve fiber layer thickness and positive scotopic threshold response amplitude. However, BMSC-derived exosomes with knockdown of Argonaute2, a protein critical for miRNAs function, markedly alleviated the effects described above. This implies that BMSC-derived exosomes may exert their neuroprotective effects through miRNA-dependent mechanisms. Finally, they identified 43 miRNAs upregulated in BMSC-derived exosomes in comparison to fibroblast-derived exosomes by using RNA sequencing. In the latter study [115], they chose the genetic DBA/2J mouse as a chronic glaucoma model, and BMSC-derived exosomes were injected into the vitreous of 3-month-old DBA/2J mice once a month for 9 months. Consistent with the previous study, the delivery of BMSC-derived exosomes also exhibited significant neuroprotection for RGCs while reducing axonal damage in the optic nerve. Particularly noteworthy is that the BMSC-derived exosomes only retained the function of RGCs in 6-month-old DBA/2J mice rather than in 9 and 12 months. Further, miRNAs in exosomes are involved in a variety of pathologies and physiologies [167,168]. It has been proven that the exosomes originating from distinct cell types can be delivered intracellularly, while all cells exhibit an ability to functionally utilize the delivered miRNAs [169].

The exosomes secreted by BMSC comprise more than 150 distinct miRNAs, which can be delivered into the target cells [170]. Previous studies have shown that both the proteins and miRNAs found in exosomes can play a therapeutic role [171], while Mead et al. [30] determined that treating RGC using exosomes in a rat optic nerve crush model relies more on miRNAs than proteins. For the first time, they delivered BMSC-derived exosomes into the eye, and the cargos delivered by the exosomes successfully reached the inner layers of the retina, including the RGC, which then elicited therapeutic benefits via the miRNA-dependent mechanisms [30]. In addition, the delivery of exosomes enriched with miR-21-5p contributed to the neuroprotection against retinal ischemia-reperfusion injuries (IRIs) and was helpful in promoting the development of cell-free treatments for glaucoma [108]. Furthermore, BMSC-derived exosomes promoted axonal growth in primary adult rat cortical neurons, which mainly relied on the action of the miRNAs in the exosomes [172]. Thus, evaluating whether exosomes can provide long-term neuroprotection in glaucoma models will be essential to apply exosomes in the future clinically.

### 3.6. Research Frontiers

In this study, there were a total of 59 keywords with a minimum of five occurrences that were identified and divided into five clusters (Figure 6A). The results reveal that some terms, such as glaucoma, microRNA, expression, open-angle glaucoma, apoptosis, aqueous humor, and retinal ganglion cell, were more frequently used terminologies in the literature (Figure 6B) and could provide some clues for the future direction of miRNAs in glaucoma research. Special attention should be paid to circRNAs, exosomes, EVs, and autophagy since they were shown as emerging frontiers in this field over the past 15 years. The following topics might merit more in-depth study in the future: (1) the roles of circRNAs in retinal neurodegeneration. (2) The neuroprotective function of exosome-derived or EVs-derived miRNAs in eye diseases. (3) Autophagy engaging in the pathogenic mechanisms of glaucoma.

### 3.7. Limitations

We used bibliometric and visualized analyses to formulate the research actuality in the field of miRNAs in glaucoma research, which enabled this research to be comparatively exhaustive and objective. Nevertheless, this research still contains some inevitable limitations. Firstly, the literature data in this research were all collected from the WoSCC database, which is the most widely used and authoritative database; however, there are still some publications that are not incorporated into the WoSCC database. Secondly, publications not written in English and papers published before 2007 and in 2023 were not adopted in this research. In addition, only original articles and reviews were analyzed in this research. Thirdly, we ignored the value of the publications, meaning that high-quality publications and low-quality publications were of similar weight. Finally, by only considering the institution of the first author, contributions from other countries in the worldwide research network may have been overlooked, especially where the main authors were from other countries.

## 4. Materials and Methods

### 4.1. Search Strategy and Data Collection

We conducted an advanced search of the WoSCC database on 28 December 2022, and used the following search terms to identify publications primarily concerning miRNAs: TS = (“microRNA*” OR “miRNA*” OR “miR”) AND TS = (“glaucoma”). We limited the search period to between 1 January 2007 and 28 December 2022. Then, the document type was limited to original and review articles, while the publication language was limited to English. The exclusion criteria are as follows. The types of publications are meeting abstract, editorial material, correction, book chapter, letter, and non-English articles. Firstly, R.Z. and Y.T. searched and screened the publications separately; however, any appearance of a problem was discussed, and a consensus was reached. The identified publications that met the inclusion criteria were exported as plain text files in the format of “Full Record and Cited References”.

### 4.2. Data Analysis

The publications were imported into VOSviewer (version 1.6.11; Leiden University, Leiden, Netherlands) and CiteSpace (Version 6.2.R4, Drexel University, Philadelphia, PA, USA) to retrieve the title, keywords, authors, institutions, countries or regions, journals, publication year, citations, average citations, and cited references. The corresponding bibliometric parameters were exported to Microsoft Excel 2010 (Redmond, Washington, WA, USA) to identify the publication trend, the distribution of document types, and the largest contributors, including prolific authors, institutions, countries or regions, and journals. VOSviewer was used to illustrate the map and depict the strength of the collaborations between authors, institutions, countries, and journals to demonstrate their scientific influence in this field. Lastly, the co-occurrence of author keywords in VOSviewer, keywords with the strongest citation bursts, and co-cited references in CiteSpace were utilized to visualize the knowledge evolution, hot topics, and potential research frontiers in this field. For the network visualization maps generated by VOSviewer, the color of the node represents clustering, the size of the node shows the number of publications or the frequency of keywords, the link between the nodes indicates the cooperative or co-occurrence relationship, and the thickness of the link suggests the strength. In the overlay visualization map produced by VOSviewer, the difference is that the color of the node implies the average publication year, purple means early, and yellow means recent [173]. For the maps generated maps in CiteSpace, there is a blue line that indicates the period, and a red line that represents the period of the bursts [174].

## 5. Conclusions

In conclusion, the number of annual publications on miRNAs in glaucoma research exhibited a continuous upward trend during the past two decades. China is a pioneering country in this field and has contributed to the development of miRNAs in glaucoma research. Of course, institutional and individual cooperation is critical to the productivity of research involving miRNAs in glaucoma and will form the backbone of any future research. The results of this research can provide the foundations and new frontiers for future research studies on miRNAs in glaucoma by summarizing and visualizing the publication trends, research hotspots, collaboration relationships, research frontiers, etc., which will enable readers to rapidly and efficiently access the useful information in this field. These findings will help the research community to explore the emerging topics and mechanisms, and provide guidance for clinical trials on glaucoma in the future.

## Figures and Tables

**Figure 1 ijms-24-15377-f001:**
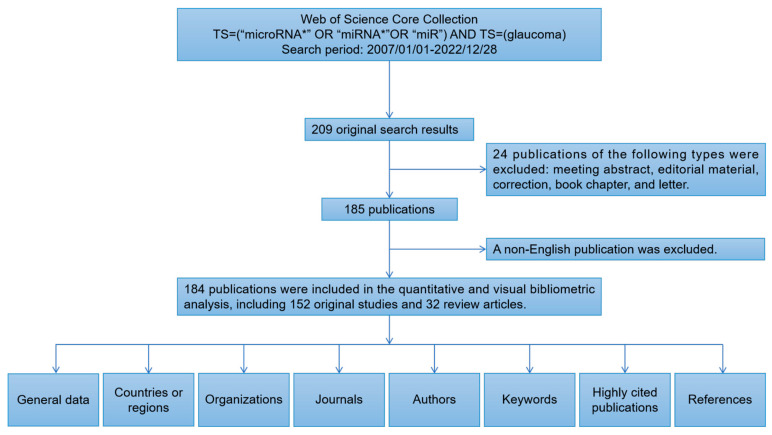
Data screening flow chart and steps of bibliometric analysis. “*” indicates the truncated version of the term (microRNA* represents microRNA, microRNAs) recognized by the search algorithm.

**Figure 2 ijms-24-15377-f002:**
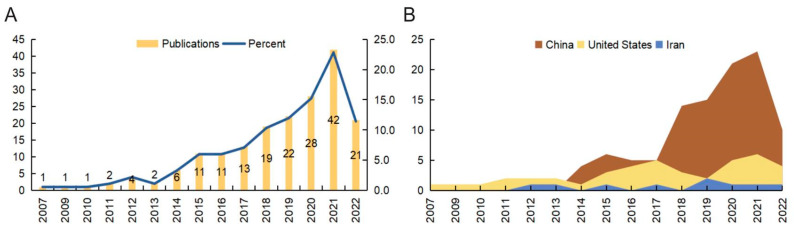
The analysis of general data. (**A**) The number of papers published each year. (**B**) The top three active countries’ annual publications.

**Figure 3 ijms-24-15377-f003:**
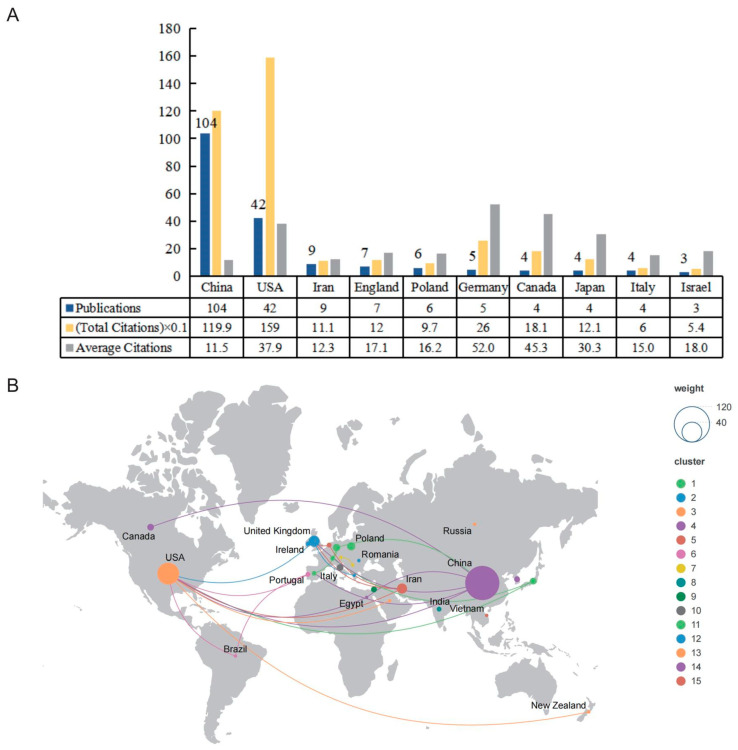
The top 10 active countries/regions and cooperative relationships in this field. (**A**) The number of publications, total citations, and average citations in the top 10 countries/regions. (**B**) The cooperative relationships of countries/regions.

**Figure 4 ijms-24-15377-f004:**
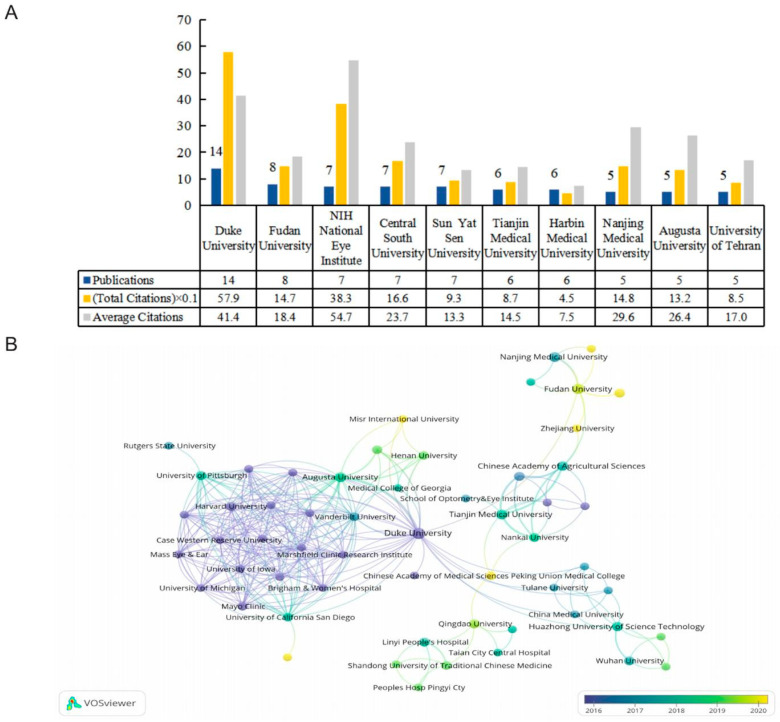
(**A**) The numbers of publications, citations, and average citations in the top 10 active institutions. (**B**) The cooperative network overlay visualization map of institutions.

**Figure 5 ijms-24-15377-f005:**
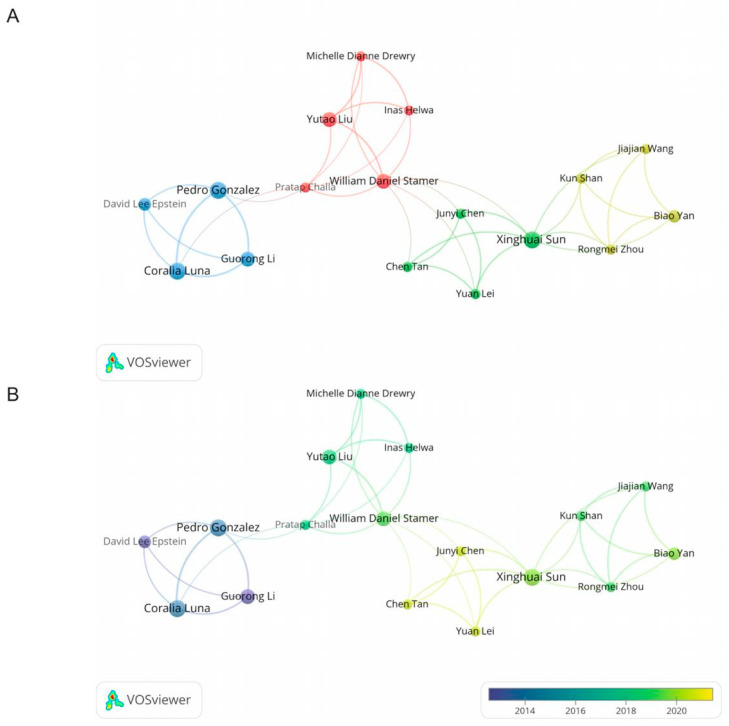
The cooperative relationships between authors. (**A**) The co-authorship network visualization map of authors related to this field. (**B**) The overlay visualization map between authors.

**Figure 6 ijms-24-15377-f006:**
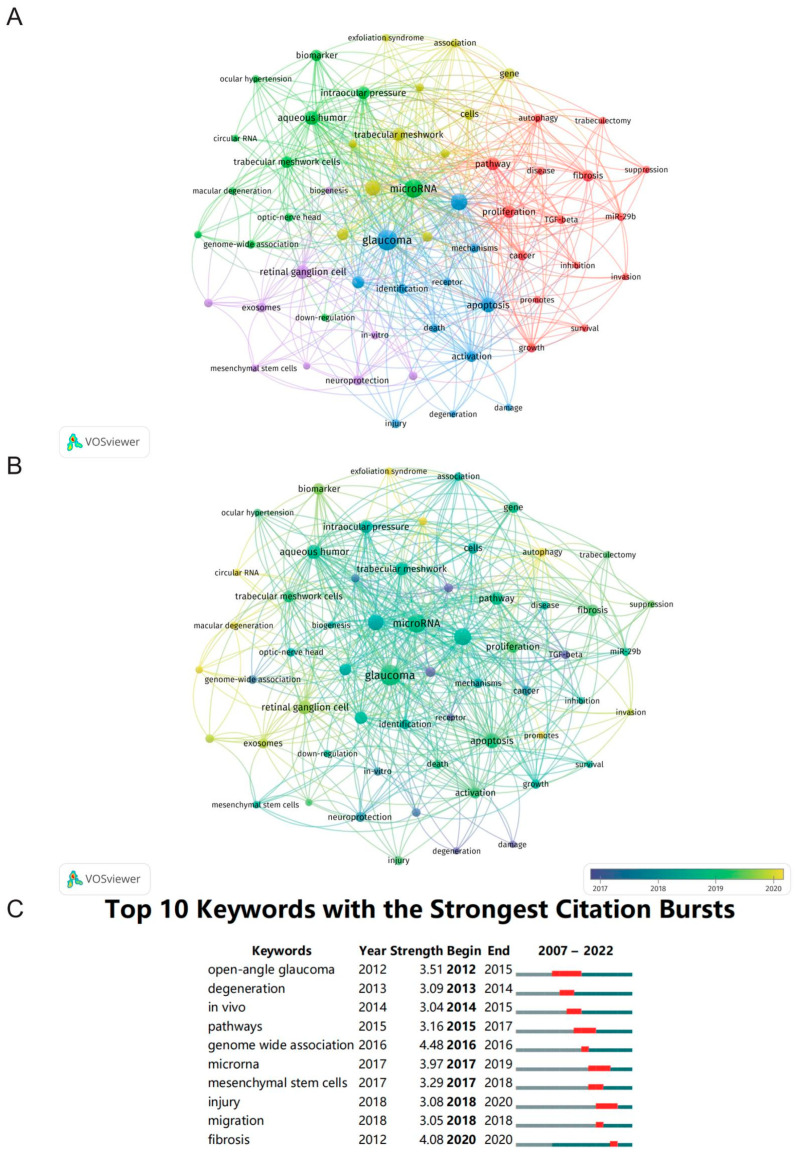
Analysis of keywords. (**A**) The co-occurrence network visualization map of keywords. (**B**) The overlay map of keywords. (**C**) The top 10 keywords with the strongest citation bursts. The red segment of the blue line represents the burst duration of a keyword.

**Table 1 ijms-24-15377-t001:** Top 10 prolific journals and co-cited journals on the application of miRNAs in glaucoma research.

Rank	Journal	Publications	Citations	Average Citations	IF (2022)	Co-Cited Journal	Co-Citations	IF (2022)
1	*Investigative Ophthalmology and Visual Science*	23	750	32.6	4.4	*Investigative Ophthalmology and Visual Science*	1107	4.4
2	*Experimental Eye Research*	10	211	21.1	3.4	*Experimental Eye Research*	414	3.4
3	*Scientific Reports*	6	115	19.2	4.6	*PloS One*	332	3.7
4	*Molecular Medicine Reports*	5	56	11.2	3.4	*Molecular Vision*	249	2.2
5	*International Journal of Molecular Sciences*	5	16	3.2	5.6	*Scientific Reports*	197	4.6
6	*Journal of Cellular Physiology*	4	107	26.8	5.6	*Journal of Biological Chemistry*	190	4.8
7	*Biomedicine and Pharmacotherapy*	4	86	21.5	7.5	*Proceedings of the National Academy of Sciences of the United States of America*	175	11.1
8	*International Journal of Molecular Medicine*	4	25	6.3	5.4	*Progress in Retinal and Eye Research*	151	17.8
9	*Molecular Vision*	3	157	52.3	2.2	*Nucleic Acids Research*	138	14.9
10	*PloS One*	3	100	33.3	3.7	*Human Molecular Genetics*	137	3.5

**Table 2 ijms-24-15377-t002:** Top 10 prolific authors and co-cited authors on the application of miRNAs in glaucoma research.

Rank	Author	Publications	Citations	Average Citations	Country	Co-Cited Author	Co-Citations	Country
1	Pedro Gonzalez	6	300	50.0	USA	Coralia Luna	109	USA
2	Coralia Luna	6	300	50.0	USA	Harry A.Quigley	56	USA
3	Xinghuai Sun	6	117	19.5	China	Ben Mead	54	Wales
4	Guorong Li	5	297	59.4	China	Guorong Li	50	China
5	Yutao Liu	5	177	35.4	USA	Hari Jayaram	46	UK
6	William Daniel Stamer	5	145	29.0	USA	Yutao Liu	39	USA
7	David Lee Epstein	4	326	81.5	USA	Yuji Tanaka	38	Japan
8	Biao Yan	4	91	22.8	China	Robert N.Weinreb	35	USA
9	Pratap Challa	3	126	42.0	USA	Guadalupe Villarreal	34	USA
10	Michelle Dianne Drewry	3	97	32.3	USA	Rudolf Fuchshofer	30	Germany

## Data Availability

All data used in this manuscript are available on the Web of Science.

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
