# Peer review of "The Application of MicroRNAs in Glaucoma Research: A Bibliometric and Visualized Analysis"

_ijms, 2023, doi:10.3390/ijms242015377_

Round 1
Reviewer 1 Report
A bibliometric study has little relevance for a clinician that seeks to understand novel approaches to the pathogenesis of glaucoma (but it may be relevant for the researcher trying to find an interesting field). Fortunately, the paper also includes a well written review of the relevant literature concerning this topic (the discussions, pages 11 to 17).
There are, however, some inadvertencies:
line 240 - I believe this is an unsupported suggestion
line 310 "aberrant ocular bloodstream" - suggests some sort of malformation. The usual explanation of normal pressure glaucoma is a defect of auto-regulation of ocular blood circulation.
line 315 - the normal intraocular pressure is between 10 and 21 mmHg
line 330 - "a small percentage of patients with IOP will progress" - in fact we the talk is about Intraocular Hypertension (not just IOP): 9% of eyes with ocular hypertension will develop glaucoma
If possible, I would ask the authors to give more detail about the use of BMSL-derived exosomes in therapy, since it appears to be a very promising research
Author Response
Response to Reviewer #1 Comments
Major comments:
(1) line 240 - I believe this is an unsupported suggestion. (-line 240 - In particular, China published a large number of articles after 2017, which may be linked to the intensification of the problem of population aging in China.)
Response: Thank you for your valuable comment. In fact, we got this information from the article entitled “National and subnational prevalence and burden of glaucoma in China: A systematic analysis” (PMID: 29302324), but we forgot to add the reference. In this article, it suggested that “Together with an ageing demography, glaucoma, especially primary glaucoma, will place an ever-increasing burden on the already stretched health care services in China, unless proactive preventive strategies are put in place.”. Based on the above information, we speculated that there may be a connection between the two. We have added the reference and revised it to: “which may be linked to the intensification of the problem of population aging in China [31].”. (line 227)
(2) line 310 "aberrant ocular bloodstream" -suggests some sort of malformation. The usual explanation of normal pressure glaucoma is a defect of auto-regulation of ocular blood circulation.
Response: Thank you for your valuable comment. Considering your comment, we revised it to: “Some other factors, such as autoimmunity, defects in auto-regulation of ocular blood circulation, low intracranial pressure…”. (lines 296-297)
(3) line 315 - the normal intraocular pressure is between 10 and 21 mmHg.
Response: Thank you for your professional advice. We have revised it and cited a new literature (PMID: 35680965). We revised it to: “The normal IOP values in healthy individuals range from 10 to 21 mmHg [64].”. (lines 301-302)
(4) line 330 - "a small percentage of patients with IOP will progress" - in fact we the talk is about Intraocular Hypertension (not just IOP): 9% of eyes with ocular hypertension will develop glaucoma.
Response: Thank you for your suggestion. I am sorry for our inaccurate description. It has been revised to: “Although intraocular hypertension is frequently observed in patients with OAGs, only a small percentage of patients with ocular hypertension will develop this disease.”. (lines 315-317)
(5) If possible, I would ask the authors to give more detail about the use of BMSC-derived exosomes in therapy, since it appears to be a very promising research.
Response: Thank you for your nice suggestion. Our analysis included 184 articles; among these, there are 4 articles focusing on BMSC-derived exosomes and glaucoma. They are: 1) Mesenchymal Stem Cell-Derived Small Extracellular Vesicles Promote Neuroprotection in Rodent Models of Glaucoma (PMID: 29392316). 2) Mesenchymal Stem Cell-Derived Small Extracellular Vesicles Promote Neuroprotection in a Genetic DBA/2J Mouse Model of Glaucoma (PMID: 30452601). 3) Bone Marrow-Derived Mesenchymal Stem Cells-Derived Exosomes Promote Survival of Retinal Ganglion Cells Through miRNA-Dependent Mechanisms (PMID: 28198592). 4) Exosomes Derived from Mesenchymal Stromal Cells Promote Axonal Growth of Cortical Neurons (PMID: 26993303). We supplemented some details about the use of BMSC-derived exosomes in therapy in section 3.5:
Mead et al. demonstrated that the therapy effects of exosomes were at least partially attributable to their miRNAs in both the glaucoma rodent models [152] and genetic DBA/2J mouse models [104]. In the former study [152], they used two glaucoma rodent models: induction of ocular hypertension with intracameral microbeads and induction of ocular hypertension with laser photocoagulation. BMSC-derived exosomes were injected into the vitreous every week or every month, and the results showed that BMSC-derived exosomes provided significant neuroprotection to RGCs while preserving retinal nerve fiber layer thickness and positive scotopic threshold response amplitude. However, BMSC-derived exosomes with knockdown of Argonaute2, a protein critical for miRNAs function, markedly alleviated the above effects. This implied that BMSC-derived exosomes may exert their neuroprotective effects through miRNA-dependent mechanisms. Finally, they identified 43 miRNAs upregulated in BMSC-derived exosomes in comparison to fibroblast-derived exosomes by using RNA Sequencing. In the latter study [104], they chose the genetic DBA/2J mouse as a chronic glaucoma model and BMSC-derived exosomes were injected into the vitreous of 3-month-old DBA/2J mice once a month for 9 months. Consistent with the previous study, the delivery of BMSC-derived exosomes also exhibited significant neuroprotection for RGCs while reducing axonal damage in the optic nerve. Particularly noteworthy is that BMSC-derived exosomes only retained the function of RGCs in 6-month-old DBA/2J mice rather than in 9 and 12 months. (lines 489-507)
while Mead et al. [161] determined that treating RGC using exosomes in a rat optic nerve crush model relies more on miRNAs than proteins. (lines 513-514)
However, it is regrettable that there is no clinical application of exosomes now; thus, evaluating whether exosomes can provide long-term neuroprotection in glaucoma models will be essential to apply exosomes in the future clinically. (lines 523-524)
Reviewer 2 Report
Zhang et al has reviewed comprehensively the role of miRNA in Glaucoma. The review is well written and should be of importance for the field. However, an in-depth review of mouse models of glaucoma utilizing miRNAs should be mentioned. Mouse models provide important mechanistic ideas for disease pathogenesis and reviewing many such contributions is somewhat lacking in this manuscript.
Author Response
Response to Reviewer #2 Comments
Major comments:
(1) Zhang et al has reviewed comprehensively the role of miRNA in Glaucoma. The review is well written and should be of importance for the field. However, an in-depth review of mouse models of glaucoma utilizing miRNAs should be mentioned. Mouse models provide important mechanistic ideas for disease pathogenesis and reviewing many such contributions is somewhat lacking in this manuscript.
Response: Thank you for your professional advice. Regarding the significance of glaucoma mouse models for providing important mechanistic ideas to disease pathogenesis, we analyzed and summarized the included articles that used glaucoma mouse models. Then, we added an in-depth review of mouse models of glaucoma utilizing miRNAs in section 3.3.5:
It is possible to explore the pathogenesis of human diseases using mouse models, which provide powerful research tools [95]. Regarding the significance of glaucoma mouse models for providing important mechanistic ideas to glaucoma pathogenesis, we analyzed and summarized the included articles that used glaucoma mouse models. In research that used acute glaucoma mouse models, in which the anterior chamber was cannulated with a needle, intravitreal infusion of polydopamine-polyethylenimine nanoparticles carried miR-21-5p [96], and MSC-exosomes involved miR-21a-5p [97,98] were used to provide new therapeutic pathways for neuroprotection against glaucoma. Moreover, there are some chronic glaucoma mouse models, which were established by episcleral venous occlusion with cauterization [99], fixing a plastic ring to the ocular equator [100], infusion of microbeads into the anterior chamber [101,102], laser photocoagulation [103] or directly using a genetic DBA/2J mouse model of glaucoma [104]. In these studies, ad-ministration of microRNA mimic/inhibitor [99-101,103], application of microRNA knockout mice [102], and injection of BMSC-derived exosomes involved microRNAs [104] were used to develop microRNA-based advanced therapies. Furthermore, some other glaucoma models, such as optic nerve crush [105] and injection of N‐methyl‐D‐aspartic acid into vitreous cavity [106], also have been applied in glaucoma researches. (lines 378-396)
(2) In addition, we have optimized the description of the methods and the presentation of results in the manuscript. (lines 89-90, 95-99, 104, 108, 123-130, 140-146, 151-152, 155-158, 172-185, 191-207, 212, 222-227, 558-560, 566-583)
(3) We have optimized and changed the Figure 2 accordingly.
